# High Expression of SOD2 Protein Is a Strong Prognostic Factor for Stage IIIB Squamous Cell Cervical Carcinoma

**DOI:** 10.3390/antiox10050724

**Published:** 2021-05-05

**Authors:** Maria Cecília Ramiro Talarico, Rafaella Almeida Lima Nunes, Gabriela Ávila Fernandes Silva, Larissa Bastos Eloy da Costa, Marcella Regina Cardoso, Sérgio Carlos Barros Esteves, Luis Otávio Zanatta Sarian, Luiz Carlos Zeferino, Lara Termini

**Affiliations:** 1Department of Obstetrics and Gynecology, Division of Gynecologic and Breast Oncology, School of Medical Sciences, State University of Campinas (UNICAMP—Universidade Estadual de Campinas), Campinas, São Paulo 13081970, Brazil; mcecilia_r@hotmail.com (M.C.R.T.); mrcardoso@mgh.harvard.edu (M.R.C.); sarian@unicamp.br (L.O.Z.S.); 2Centro de Investigação Translacional em Oncologia, Instituto do Câncer do Estado de São Paulo, Hospital das Clínicas da Faculdade de Medicina da Universidade de São Paulo, São Paulo 05403911, Brazil; rafaellaln30@gmail.com (R.A.L.N.); gabriela.avila@hc.fm.usp.br (G.Á.F.S.); 3Department of Pathology, State University of Campinas (UNICAMP—Universidade Estadual de Campinas), Campinas, São Paulo 13083887, Brazil; lbeloy@unicamp.br; 4Department of Radiotherapy, Division of Gynecologic and Breast Oncology, Women’s Hospital Professor Doutor José Aristodemo Pinotti—Centro de Atenção Integral à Saúde da Mulher (CAISM), State University of Campinas (UNICAMP—Universidade Estadual de Campinas), Campinas, São Paulo 13083881, Brazil; sergiocb@unicamp.br

**Keywords:** biomarkers, tumor, superoxide dismutase 2, antioxidants, uterine cervical neoplasms, survival analysis, prognosis

## Abstract

High superoxide dismutase 2 (SOD2) expression is associated with a poor prognosis at many cancer sites, the presence of metastases, and more advanced cervical cancer. This study aims to determine whether SOD2 protein expression is associated with the prognosis of stage IIIB cervical carcinoma. Methods: sixty-three patients with stage IIIB squamous cell cervical carcinoma were included. The evaluation of SOD2 expression by immunohistochemistry was based on a positive cell ratio score and the staining intensity score. Taking disease recurrence and death as endpoints, receiver operating characteristic curves were used to discriminate between high and low SOD2 expression. Results: high SOD2 expression was associated with recurrence (*p* = 0.001), distant recurrence (*p* = 0.002), and death (*p* = 0.005). A multivariate analysis showed that patients with high SOD2 expression had a threefold increased risk for recurrence (HR = 3.16; 1.33–7.51) and death (HR = 2.98; 1.20–7.40) compared with patients who had low SOD2 expression. Patients with high SOD2 expression had shorter disease-free survival (*p* = 0.001) and overall survival (*p* = 0.003) than patients with low SOD2 expression. Conclusion: high SOD2 expression is a strong prognostic factor for stage IIIB squamous cell carcinoma of the cervix and could be used as a prognostic marker in women with cervical carcinoma.

## 1. Introduction

Cervical cancer is a very common malignant neoplastic disease, which is treatable even at advanced stages and has more than 600,000 new cases diagnosed each year worldwide [1]. The development of clinical tools for the diagnosis, treatment, and follow-up of malignant neoplasms may help improve the outcome and may benefit patients.

Currently, HPV (human papillomavirus) testing, cytology, or a combination of HPV testing cytology are the major means to screen a population for cervical cancer [2]. Having a biomarker to screen cervical cancer would allow for the identification of asymptomatic women at risk of developing cancer in time to receive treatment before the occurrence of invasion [3]. From this perspective, biomarkers have emerged for complementary use in patient assessment. Broadly, biomarkers are molecules in the body, which when measured, could characterize a biological state as normal or abnormal. In oncology, biomarkers could be used to estimate a patient’s risk of developing the disease, to screen cancers in routine exams, and to determine a tumor’s malignancy. For prognosis, it could be used to monitor the disease, to detect relapses, to determine treatment response, and to evaluate a patient’s survival. In general, the use of biomarkers could support diagnostic hypotheses informing the oncologist about possible outcomes and could offer the patient a possible cure or at least the prolongation of survival [4,5,6].

The family of superoxide dismutase (SOD) metalloenzymes is very important in antioxidant defense. In mammals, three distinct isoforms of superoxide dismutase, SOD1 (cytoplasmic), SOD2 (mitochondrial), and SOD3 (extracellular), have been identified and characterized. However, of the three, SOD2 is the only isoform that is essential for the survival of aerobic organisms [7,8]. Additionally, known as manganese superoxide dismutase, the SOD2 gene is located on chromosome 6q25.3 [7,9,10].

The SOD2 gene showed a differential expression in association with inflammatory response when the profile of the global transcription of normal and HPV-immortalized keratinocytes was analyzed [11]. It is known that SOD2 protein is differentially expressed according to the severity of the cervical epithelial neoplasia. Adenocarcinoma and squamous cervical carcinoma have stronger expressions than other cervical lesions [12,13]. Infection by HPV 16 and 18 are the strongest risk factors for cervical cancer development, but the expression of SOD2 seems to be a risk factor independent of the role of the HPV infection [13]. The presence of SOD2 has also been associated with pelvic lymph node metastasis in patients with early stage cervical carcinoma [14]. However, the possible association of SOD2 expression with the prognosis of this neoplasm is still unknown. The aim of this study was to determine whether SOD2 protein expression is associated with the prognosis of stage IIIB cervical carcinoma.

## 2. Materials and Methods

### 2.1. Case Selection

This study selected patients from a cohort consisting of tumor samples from women included in the prospective clinical trial named “Concomitant cisplatin plus radiotherapy and high-dose-rate brachytherapy versus radiotherapy alone for stage IIIB epidermoid cervical cancer: a randomized controlled trial”, conducted at the Women’s Hospital Prof. Dr. José Aristodemo Pinotti, State University of Campinas (UNICAMP), Brazil, and approved by the local Research Ethics Committee (protocol No. 2.426.854). Briefly, from September 2003 to July 2010, the clinical trial included 147 women with stage IIIB cervical squamous cell carcinoma, a creatinine clearance higher than 60 mL/min/1.73 m², normal liver enzymes, a Karnofsky score higher than 70%, and baseline hemoglobin levels of at least 10 mg/dL. Of the 147 women, 72 received cisplatin concomitantly with radiotherapy (CRT) and 75 women received radiotherapy alone (RT); follow-up lasted seven years. All women received teletherapy with a dose of 45 Gy for the pelvic region in 25 fractions, 14.4 Gy of reinforcement in the compromised parametria, and high-dose brachytherapy of 7 Gy in four weekly fractions. The women in the CRT group also received weekly chemotherapy with cisplatin (40 mg/m^2^) concomitant with teletherapy. Response to treatment was assessed one month after its completion. Follow-up for the treated women was performed every four months in the two years following treatment, twice at six month intervals in the third year, and once a year thereafter. Follow-up lasted until June 2013 [15].

Of the 147 women initially included in the clinical trial, 49 lacked sufficient formalin-fixed paraffin-embedded tissue for additional immunohistochemical assays and 35 women had no paraffin-embedded tissue because their diagnostic biopsies were performed in other health clinics. Thus, after careful screening, the initial study sample was downsized to 63 women (Research Ethics Committee approval CAAE# 55014816.5.0000.5404, 11 December 2017). The available samples were cut and stained with hematoxylin and eosin to verify the tissue and the tumor component quality. The immunohistochemical assay did not work for eight women. Therefore, analyses were performed in the 55 remaining women: 28 women in the CRT group and 27 in the RT group.

### 2.2. Immunohistochemistry Assay for SOD2 Detection

The immunohistochemical assays were performed at the Cancer Innovation Laboratory, Centro de Investigação Translacional em Oncologia, Instituto do Câncer do Estado de São Paulo Octavio Frias de Oliveira—ICESP, Universidade de São Paulo, Brazil, by automated reaction using the UltraView Universal DAB Detection Kit^®^ (Ventana Medical Systems, Inc., Roche, Tucson, AZ, USA) and the Ventana BenchMark GX equipment (Roche Diagnostics, Mannheim, Germany), according to the manufacturer’s instructions. In summary, the slides were submitted to an antigenic recovery process, carried out with a tris-based buffer, pH 8.0 (Ultra Cell Conditioning Solution—Roche Diagnostics, Mannheim, Germany) under heat for 30 min. Next, rabbit polyclonal antibodies directed against SOD2 protein (ab 13533, Abcam, Cambridge, MA, USA) were used in a 1:1000 concentration for 32 min. Positive reactions were visualized with a cocktail (UltraView Universal HRP Multimer—Roche Diagnostics, Mannheim, Germany) containing peroxidase conjugated anti-mouse and anti-rabbit secondary antibodies in the presence of DAB, resulting in a brown precipitate. The slides were counterstained with hematoxylin for 20 min. This entire process was performed inside the Ventana BenchMark GX equipment in a closed system. Samples of cervical squamous cell carcinomas previously tested for SOD2 expression by immunohistochemistry were used as a reaction control and were incubated with or without anti-SOD2 antibodies (data not shown).

### 2.3. Analyses of Immunohistochemical Reactions

The immunohistochemical positivity reaction was performed by a quantitative method using Image J^®^ and evaluated by an experienced pathologist blinded to clinical and pathological data [16]. Evaluation covered the ratio of stained cells to the total number of cells and staining intensity in the representative areas of squamous cell carcinoma. Staining intensity was classified as 0 (no staining), 1 (weak), 2 (moderate), or 3 (strong). Figure 1 illustrates representative immunohistochemical assays with different SOD2 expression features. Cells with granular cytoplasm staining were considered positive. Microphotographs of the three best fields at high magnification (×400) were taken by a 995 Nikon digital camera. The selected regions include areas with greater intensity and extension of cellular expression for the marker (“hotspot areas”), focusing on the component of carcinomatous cells. Using the open-source image processing software ImageJ^®^, the number of SOD2-positive cells and the total number of cells per field were assessed in each picture to obtain the positive cell ratio in each case. Subsequently, the stained cell ratio score and the staining intensity score were multiplied by each other to obtain a final score for each of the fields that were analyzed. A score average ranging between 0 and 3 was calculated for comparison analysis (Appendix A) [17,18].

Initially, the SOD2 expression scores were analyzed as a continuous variable; however, no difference was observed (Appendix A). Hence, due to the absence of clinical parameters to classify the scores as dichotomous categorical variables, receiver operating characteristic (ROC) curves were created using disease recurrence and death as the outcomes, and the best discriminating score value was 1.9 (Appendix A). Therefore, the scores were classified as low (<1.9) and high expressions (≥1.9) for analytical purposes. Despite the small sample size, the analysis of the ROC curve as a predictor of recurrence showed a trend towards significance (*p* = 0.09).

### 2.4. Statistical Analysis

Frequency tables were compiled to present the sample characteristics. A comparison of the categorical variables between groups was performed using the chi-square test or Fisher’s exact test. A comparison of numerical variables between groups was carried out using the Mann–Whitney test due to the absence of a normal distribution of the variables. The analyses of the DFS (disease-free survival/recurrence) and of the OS (overall survival/death) curves were performed using the Kaplan–Meier method and the log-rank test. Factors associated with survival were analyzed by the Cox proportional hazards regression. The significance level was set at 5% (*p* < 0.05).

## 3. Results

The mean age of the patients was 53.9 years (24.0–76.0), the mean duration of treatment was 91 days (58–184), and the mean follow-up time was 45.8 months (5.4–90.9). Treatment duration was similar for women treated with RT or with CRT (no statistically significant difference). Table 1 shows the SOD2 protein expressions according to patient’s age, pathological tumor grade, treatment received, and toxicity. There were no statistically significant associations between the variables under analysis.

Table 2 shows SOD2 protein expressions in relation to disease recurrence and death. A total of 78.6% of the patients whose carcinomas showed high SOD2 expression had recurrences, while only 29.3% of patients with carcinomas expressed low SOD2 levels. The difference was statistically significant (*p* = 0.001). Furthermore, 71.4% of the patients who had high SOD2 expression died while the same outcome was observed in only 29.3% of the patients with low expression (*p* = 0.005). In addition, 57.1% of the patients with high SOD2 expression showed distant recurrence, which occurred in only 12.2% of the patients with low expression. This is evidence that a high expression is associated with systemic relapse (*p* = 0.002).

Table 3 shows the Cox regression results from the analysis of DFS and OS. A univariate analysis revealed that, when samples presented high SOD2 expressions, the patients showed higher risk for recurrence (HR = 3.68, 95% CI = 1.58–8.57) and for death (HR = 3.54; 95% CI = 1.46–8.58) than patients with low SOD2 expression. In a multivariate analysis, high expression exhibited statistically significant findings: patients whose tumors had a high expression had higher risks for recurrence (HR = 3.16, 95% CI = 1.33–7.51) and for death (HR = 2.98; 95% CI = 1.20–7.40) than patients who had a low expression.

Figure 2 shows the Kaplan–Meier curves for DFS and OS. According to the log-rank test, women with shorter DFS had carcinomas with higher SOD2 expression (Figure 2a) (*p* = 0.001). In the analysis of OS, women whose carcinomas had high SOD2 expressions also presented poor OS (Figure 2b) (*p* = 0.003).

## 4. Discussion

The univariate analysis of data from our study showed that high SOD2 expression was associated with reduced disease-free survival (DFS) and overall survival (OS) in patients with cervical cancer. A multivariate analysis, including clinicopathological parameters such as age, anatomopathological tumor grade, type of treatment, and SOD2 expression, revealed that SOD2 expression was an independent prognostic factor. Additionally, the Kaplan–Meier curves for DFS and OS demonstrated a difference in prognosis between low and high SOD2 expression, since most of the women with high SOD2 expressions relapsed and died during the first three years of follow-up. These findings add to the growing body of evidence that supports the association between SOD2 protein expression and cancer behavior.

Evidence indicates that SOD2 expression in neoplastic lesions of the cervical epithelium increases with the severity of the disease [12,13]. Moreover, although the main cause of cervical cancer development is persistent high-risk HPV infections, there is evidence that SOD2 expression may play a role in the carcinogenesis of cervical epithelium independently of the type of HPV [13]. A microarray approach found SOD2 to be differentially expressed in a gene expression signature and associated with pelvic lymph node metastasis in cervical cancer. Furthermore, multivariate analysis indicated that the gene signature was a predictor of pelvic lymph node metastasis in the primary tumor [14]. In summary, SOD2 protein expression level seems to play a role in cervical carcinogenesis and, more specifically, to be involved in the progression of the disease.

Studies conducted during the last decades indicate that SOD2 plays a dual role in cancer. This protein may exhibit both anti-tumoral as well as pro-tumoral roles depending on tumor type, tumor stage, and cellular context. In fact, the function of SOD2 in the development and progression of cancer is still poorly understood. This protein may act as a prooxidant, collaborating in the induction and promotion of cancer through the accumulating hydrogen peroxide. On the other hand, SOD2 may act as a protective antioxidant that reduces superoxide anion [19,20,21]. In general, studies conducted in vitro tended to demonstrate that SOD2 had a protective role against the malignant phenotype. A high expression of SOD2 was related to increased cell differentiation, decreased cell growth or proliferation, and reversion of the malignant phenotype to a nonmalignant phenotype [22,23].

In vivo studies have provided evidence of the association between high SOD2 expression and more aggressive tumor phenotypes. Generally, in the early stages of disease, the expression of SOD2 is low, indicating that SOD2 could be associated with the onset of tumor development [24]. In contrast, SOD2 levels are commonly increased in advanced cancer tissues, especially in metastatic tumors, when compared to benign counterparts [7,24,25]. This might occur because the metabolism of the tumor cells is progressively more aberrant, leading to an accumulation of excessive amounts of ROS, which would induce further cell damage and promote tumor development [25,26,27]. To prevent cell damage, the antioxidant system is activated, increasing the expression of antioxidant agents, such as SOD2, to counterbalance the effect of oxidants [28].

Hyper-expressed SOD2 appears to combat oxidative stress, thereby preventing apoptosis. However, when this occurs in cancer cells or tumor tissue, SOD2 is a two-edged sword because antioxidant excess can induce the survival of damaged cells and then promote their proliferation and prompt the survival of neoplastic cells [29].

The regulation of SOD2 expression and activity occurs both at the transcriptional and posttranslational levels. The activity of SOD2 protein can be regulated by protein localization, interaction with other cellular factors, transition metal incorporation, oxidation, nitration, S-gutathionylation, phosphorylation, ubiquitination, and acetylation. An extensive in-depth discussion of the complex regulation of SOD2 expression and activity in cancer is beyond the scope of our study. For a detailed description of the many variables involved in SOD2 regulation in cancer, the reader is referred to several recent excellent reviews [21,24,28].

The data from our study showed that high SOD2 expression was associated with distant recurrence but not with local or regional recurrences. These findings suggest that SOD2 also has a role in the pathogenesis of metastases; this would explain the association of high SOD2 expression with poor prognosis. Our results raise the possibility of using SOD2 as a therapeutic target for malignant neoplasms. Thus, it is possible to propose an initial SOD2-modulating therapy to be followed by a more specific treatment.

In agreement with the results of this study, most studies of SOD2 expression relative to cancer showed that high expression of the protein is associated with the presence of metastases and a dire prognosis in some malignancies. Comparing frozen biopsy specimens of lung, colon, and prostate tissues, SOD2 was expressed in higher amounts during tumor progression when compared to controls [19]. The SOD2 expression was also increased in samples of esophageal tumor tissue, compared to tissues without neoplasia. In addition, the mean OS of patients who had high SOD2 expression was lower than that of patients with low expression [30]. An analysis of SOD2 expression conducted in several tumor samples including lymphoma [31,32]; glioblastoma [33]; and bladder [34], colorectal [35], oral [36,37,38], lung [39], kidney [40,41], ovarian [42,43], penile [44], and breast cancers [45] have often associated its upregulation with metastasis and poor disease outcome. The observations described above indicate that increased SOD2 expression is a common trait of different tumor types. However, the results presented in our study clearly indicate that the levels of expression of this protein are associated with DFS and OS in patients with stage IIIB squamous cell carcinoma of the cervix, suggesting that SOD2 expression may be considered a biomarker for cervical disease outcome.

The small sample size is a weakness of this study because it prevented the analysis of more variables, such as tumor grade and other prognostic factors. However, this study showed a clear statistical association between SOD2 expression and the prognosis of cervical cancer, which strongly suggests the existence of a relevant biological event. Nevertheless, new studies with a larger number of samples, ideally from different centers, are needed to further confirm the results obtained in the present work. A major strength of this study is the inclusion of women with advanced cervical carcinomas. These patients have a high probability of recurrence and death and are therefore sensitive to prognostic analyses. It should also be noted that cases were homogeneous, including tumors at the same stage, for the same histological type. Moreover, patients were treated and followed-up at the same institution. Our approach allowed us to predict a clinical outcome using a small tissue sample at the time of diagnosis using the basic method of immunohistochemistry, widely available at diagnostic centers.

Finally, two relevant questions emerged from this study. The first is whether SOD2 expression is associated with the prognosis for earlier invasive cervical cancer as it is in more advanced cases. The second, considering that SOD2 expression is associated with distant recurrence and death, is whether it makes sense to consider more aggressive therapy in patients with higher SOD2 expression in an attempt to improve the prognosis of such cases. Further studies are needed to determine the clinical implication of this factor.

## 5. Conclusions

In conclusion, this study showed that a high SOD2 expression is associated with a very pessimistic prognosis for squamous cell carcinoma of the uterine cervix IIIB. These findings suggest that SOD2 protein could be considered a candidate for a prognostic marker in clinical approaches for women with cervical carcinoma and, together with the evidence available in the literature, suggest that the oxidative system, especially SOD2, could be considered therapeutic targets for malignant neoplasms.

## Figures and Tables

**Figure 1 antioxidants-10-00724-f001:**
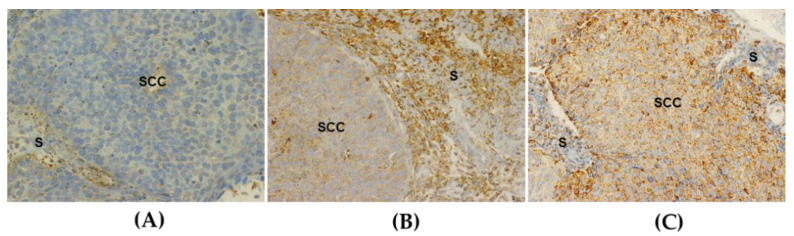
Representative immunohistochemical analysis of SOD2 expression in squamous cell carcinoma of the cervix (200×). Granulated-like cytoplasmic expression in negative (**A**), moderate (**B**), and strong (**C**) staining in neoplastic cells. SCC = squamous cervical carcinoma. S = stromal cells.

**Figure 2 antioxidants-10-00724-f002:**
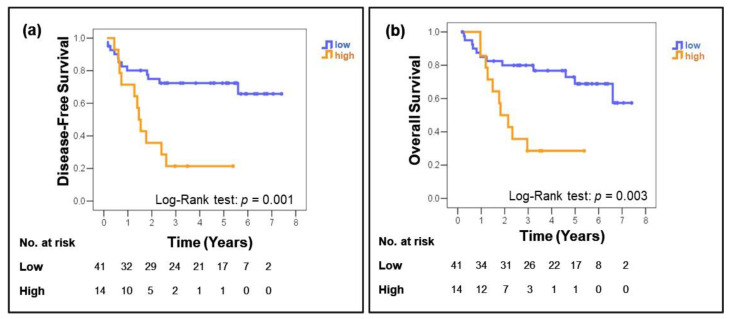
Disease-free (**a**) and overall (**b**) survival rates of the groups categorized by superoxide dismutase (SOD2) expression in a cohort of patients with squamous cervical cancer. SOD2 expression was translated into a score based on the ratio of stained cells and staining intensity and then classified as low (<1.9) or high expression (≥1.9). A high expression of SOD2 was associated with a trend toward a shorter disease-free survival (*p* = 0.001) and was correlated significantly with poor overall survival (*p* = 0.003) in squamous cervical cancer.

**Table 1 antioxidants-10-00724-t001:** Association between SOD2 protein and clinicopathological characteristics in cervical cancer IIIB.

		SOD2 Expression ^α^		*p*-Value ^†^
	*n*	Low (*n* = 41)	High (*n* = 14)	
Age				
<50	15	10 (24.4%)	5 (35.7%)	0.395
50–59	20	17 (41.5%)	3 (21.4%)	
≥60	20	14 (34.2%)	6 (42.9%)	
Anatomopathological Tumor Grade *				
1–2	45	34 (85.0%)	11 (78.6%)	0.681
3	9	6 (15.0%)	3 (21.4%)	
Treatment				
CRT	28	22 (53.7%)	6 (42.9%)	0.485
RT	27	19 (46.3%)	8 (57.1%)	
Acute Toxicity *				
Yes	15	13 (34.2%)	2 (15.4%)	0.297
No	36	25 (65.8%)	11 (84.6%)	
Late Toxicity *				
Yes	23	19 (46.3%)	4 (30.8%)	0.323
No	31	22 (53.7%)	9 (69.2%)	

CRT = chemoradiotherapy. RT = radiotherapy. * Some cases lacked information on tumor grade, acute toxicity, and late toxicity; therefore, only cases with such information are presented. ^α^ SOD2 expression was translated into a score based on the ratio of stained cells and on staining intensity and then classified as low (<1.9) or high expression (≥1.9). ^†^ Chi-square test and Fisher’s exact test.

**Table 2 antioxidants-10-00724-t002:** Expression of SOD2 protein according to recurrence, local recurrence, regional recurrence, distant recurrence, and death.

	*n*	SOD2 Expression ^α^		*p*-Value ^†^
		Low (*n* = 41)	High (*n* = 14)	
Recurrence				
Yes	23	12 (29.3%)	11 (78.6%)	**0.001**
No	32	29 (70.7%)	3 (21.4%)	
Local recurrence				
Yes	11	7 (17.1%)	4 (28.6%)	0.443
No	44	34 (82.9%)	10 (71.4%)	
Regional recurrence				
Yes	10	6 (14.6%)	4 (28.6%)	0.255
No	45	35 (85.4%)	10 (71.4%)	
Distant recurrence				
Yes	13	5 (12.2%)	8 (57.1%)	**0.002**
No	42	36 (87.8%)	6 (42.9%)	
Death				
Yes	22	12 (29.3%)	10 (71.4%)	**0.005**
No	33	29 (70.7%)	4 (28.6%)	

^α^ SOD2 expression was translated into a score based on the ratio of stained cells and on staining intensity and then classified as low (<1.9) or high expression (≥1.9). ^†^ Chi-square test and Fisher’s exact test.

**Table 3 antioxidants-10-00724-t003:** Univariate and multivariate Cox regression analyses for disease-free survival and overall survival according to patient’s age, tumor grade, treatment, and SOD2 expression (*n* = 55).

Univariate Analysis	Categories	DFS		OS	
		HR ^#^	95% CI ^#^	HR ^##^	95% CI ^##^
Age	<50	2.34	0.81–6.79	3.88	1.16–12.97
	50–59	1.00	-	1.00	-
	≥60	1.66	0.59–4.68	2.67	0.84–8.51
Anatomopathological tumor grade *	1 and 2	1.00	-	1.00	-
	3	1.14	0.39–3.36	1.17	0.39–3.47
Treatment	CRT	1.00	-	1.00	-
	RT	1.57	0.69–3.58	1.35	0.58–3.13
SOD2 expression ^α^	Low	1.00	-	1.00	-
	High	3.68	1.58–8.57	3.54	1.46–8.58
**Multivariate Analysis**					
Age	<50	2.62	0.78–8.85	5.07	1.24–20.75
	50–59	1.00	-	1.00	-
	≥60	1.26	0.44–3.61	2.15	0.65–7.04
Anatomopathological tumor grade *	1 and 2	1.00	-	1.00	-
	3	1.36	0.42–4.37	1.61	0.48–5.40
Treatment	CRT	1.00	-	1.00	-
	RT	1.82	0.72–4.61	1.85	0.69–4.94
SOD2 expression ^α^	Low	1.00	-	1.00	-
	High	3.16	1.33–7.51	2.98	1.20–7.40

DFS = disease-free survival. OS = overall survival. ^#^ HR (hazard ratio) = hazard ratio to relapse, with censored = 32 patients and relapse = 23 patients. ^##^ HR (hazard ratio) = hazard ratio to death, with censored = 33 patients and death = 22 patients. CI = confidence interval. * Some cases lacked information on tumor grade, acute toxicity, and late toxicity; therefore, only cases with such information are presented. ^α^ SOD2 expression was translated into a score based on the ratio of stained cells and on staining intensity and then classified as low (<1.9) or high expression (≥1.9).

## Data Availability

The data presented in this study are available in Appendix A. Other data presented in this study are available on request from the corresponding author. The data are not publicly available due to ethical restrictions.

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
