# Peer review of "High Expression of SOD2 Protein Is a Strong Prognostic Factor for Stage IIIB Squamous Cell Cervical Carcinoma"

_antioxidants, 2021, doi:10.3390/antiox10050724_

Round 1

Reviewer 1 Report

Talarico and colleagues have utlised data from a clinical trial for their sub-analysis. The topic is of interest but there are some methodological and analytical issues with the current version which need to be addressed.

Major

- The authors make claims about the association with SOD2 expression with cancer recurrence etc, however their own data shows that higher levels of SOD2 are associated (if not statistically significantly) with age over 60 years, Grade of tumour, Treatment, and lower levels of toxicity. A statistical examination of the 23 recurrent cases identified in Table 2 in relation to age, and tumour type (at a minimum) would be critical prior to any claims about SOD2 as its expression may be part of a cohort effect rather than causational as may be implied from the authors analysis.

- The examination of the histology samples has flaws in that it does not appear that the visualization, or examination were blinded which may introduce a bias. There also appears to be a lack of a staining control for the histological sections to demonstrate that staining is equal across different sections.

- It is unclear where the method describe for SOD2 analysis (e.g. the data that leads to <1.9 etc) comes from as it does not appear to be a previously published method.

- the methods lack enough information to be reproducible, and an example the only data available on the staining for SOD2 the authors mention the Ventana and the antibody but no staining conditions or other features required.

Minor

- There presentation of the data as percentages in Table 1 and 2 is difficult to understand with raw numbers being more benificial

- The images in Figure 1 are at x200 but the methods suggest that images were taken at x400

- There does not appear to be any cross staining to identify the SCC which makes the images in Figure 1 difficult to interpret

  • the manuscript as a whole is way to verbose and includes much too much basic information, e.g. the entire paragraph at the top of page 6 of 12 is a very very basic overview of SOD2 and ROS and it is not necessary.

Author Response

Dear Editor and Reviewers,

Coauthors and I very much appreciated these encouraging, critical and constructive comments and suggestions provided by reviewers on this manuscript, and we strongly believe these have contributed to increase the scientific value and to communicate more clearly. We are submitting the corrected manuscript with changes incorporated the manuscript. These changes aim to address comments given by reviewers. Additionally, we have added notes below to each comment provided:

Once again, we thank you for your time spent to review our paper.

We look forward to meet your expectations.

-The authors

Reviewer #1:

“Talarico and colleagues have utlised data from a clinical trial for their sub-analysis. The topic is of interest but there are some methodological and analytical issues with the current version which need to be addressed.”

We wish to deeply thank the Reviewer #1 for the careful and constructive revision of the manuscript. Below there is a point-by-point answer to issues raised by the Reviewer.

“The authors make claims about the association with SOD2 expression with cancer recurrence etc, however their own data shows that higher levels of SOD2 are associated (if not statistically significantly) with age over 60 years, Grade of tumour, Treatment, and lower levels of toxicity. A statistical examination of the 23 recurrent cases identified in Table 2 in relation to age, and tumour type (at a minimum) would be critical prior to any claims about SOD2 as its expression may be part of a cohort effect rather than causational as may be implied from the authors analysis.”

The point raised by Reviewer #1 is interesting and needs to be made perfectly clear. The data analyses suggested that there is no cohort effect of the SOD2 expression. In fact, no direct association was observed between age and SOD2 expression. Also, the multivariate analyses (Cox regression) showed independent association of age and SOD2 expression with overall survival. It is relevant to highlight that disease-free survival and overall survival take into account respectively disease recurrence and death in the timeline after diagnosis. Finally, according to multivariate analyzes, overall survival was worse for younger women, whereas for a cohort effect, the opposite would be expected.

“The examination of the histology samples has flaws in that it does not appear that the visualization, or examination were blinded which may introduce a bias.”

We thank Reviewer #1 for raising this concern giving us the opportunity to explain it. The professionals who performed the immunohistochemistry analyses had no information about the clinical outcome of the patients. We have included this information in the methods description (lines 130-131).

“There also appears to be a lack of a staining control for the histological sections to demonstrate that staining is equal across different sections.”

We thank Reviewer #1 for this observation. All the immunohistochemical reactions were performed using appropriate positive and negative controls. Besides, the study was performed using reactions controls where samples were incubated with or without anti-SOD2 antibody (lines 125-127).

Additionally, positive biological controls were included in the analyses, to show that the antibody used works properly. The negative controls used showed that there is no nonspecific binding of the secondary antibody in the reaction. The specific method has been rewritten to include all the details of the reaction (lines 110-127).

“It is unclear where the method describe for SOD2 analysis (e.g. the data that leads to <1.9 etc) comes from as it does not appear to be a previously published method.”

The methodology to analyze the cell immunoreactivity based in images and reported by scores has been used in Pathology (Vijayashree, Aruthra, and Ramesh Rao 2015; Da Costa, Triglia, and Andrade 2017; Barreta et al. 2019). Due to the lack of any clinical parameters to establish the cut-off of scores as dichotomous categorical variables, receiver operating characteristic (ROC) curves were built using disease recurrence and death as outcomes, and the best discriminating score value for the study was 1.9 (Content already described in the article).

The ROC curve method is a statistical analysis that have been used a lot to show the connection between clinical sensitivity and specificity for every possible cut-off for a test or a combination of tests. For this study, 1.9 was best cut-off indicate by ROC curves analyses for the study parameters.

Vijayashree R, Aruthra P, Rao KR. A comparison of manual and automated methods of quantitation of oestrogen/progesterone receptor expression in breast carcinoma. J Clin Diagn Res. 2015 Mar;9(3):EC01-5. doi: 10.7860/JCDR/2015/12432.5628. Epub 2015 Mar 1. PMID: 25954622; PMCID: PMC4413070.

Da Costa, L.B.E.; Triglia, R.D.M.; Andrade, L.A.L.D.A. P16INK4a, Cytokeratin 7, and Ki-67 as Potential Markers for Low-Grade Cervical Intraepithelial Neoplasia Progression. J. Low. Genit. Tract Dis. 2017, 21, 171–176, doi:10.1097/LGT.0000000000000310.

Barreta, A.; Sarian, L.O.; Ferracini, A.C.; Costa, L.B.E.; Mazzola, P.G.; de Angelo Andrade, L.; Derchain, S. Immunohistochemistry expression of targeted therapies biomarkers in ovarian clear cell and endometrioid carcinomas (type I) and endometriosis. Hum. Pathol. 2019, 85, 72–81, doi:10.1016/j.humpath.2018.10.028.

“the methods lack enough information to be reproducible, and an example the only data available on the staining for SOD2 the authors mention the Ventana and the antibody but no staining conditions or other features required.”

The specific method has been rewritten to include all the details of the reaction (lines 110-127) “The immunohistochemical assays were performed at the Cancer Innovation Laboratory, Centro de Investigação Translacional em Oncologia, Instituto do Câncer do Estado de São Paulo Octavio Frias de Oliveira – ICESP, Universidade de São Paulo, Brazil, by automated reaction using the UltraView Universal DAB Detection Kit ® (Ventana Medical Systems, Inc., Roche, Tucson, Arizona) and the Ventana BenchMark GX equipment (Roche Diagnostics, Mannheim Germany), according to the manufacturer’s instructions. In summary, the slides were submitted to antigenic recovery process, carried out with a tris-based buff-er, pH 8.0 (Ultra Cell Conditioning Solution - Roche Diagnostics, Mannheim Germany) under heat, for 30 minutes. Next, rabbit polyclonal antibodies directed against SOD2 protein (ab 13533, Abcam, Cambridge, USA) were used in a 1:1000 concentration, for 32 minutes. Positive reactions were visualized with a cocktail (UltraView Universal HRP Multimer - Roche Diagnostics, Mannheim Germany) containing peroxidase conjugated anti-mouse and anti-rabbit secondary antibodies in the presence of DAB, resulting in a brown precipitate. The slides were counterstained with hematoxylin for 20 minutes. This whole process occurs inside the Ventana BenchMark GX equipment, in a closed system. Samples of a cervical squamous cell carcinomas previously tested for SOD2 expression by immunohistochemistry were used as a reaction control and were incubated with or without anti-SOD2 antibody (data not shown).

“There presentation of the data as percentages in Table 1 and 2 is difficult to understand with raw numbers being more beneficial.”

We agree with Reviewer #1. In the present version of the manuscript we have included the raw number together to percentage information in tables 1 and 2.

“The images in Figure 1 are at x200 but the methods suggest that images were taken at x400”

We thank Reviewer #1 for the opportunity to make this point clear. The objective of Figure 1 was to present a representative result of the immunohistochemical reaction showing SOD2 in a general tissue view. This image, taken with lower magnification (200x) shows not only the tumor tissue but also the stroma. Some components of the stroma, particularly the inflammatory infiltrate, can stain positive for SOD2 expression. On the other hand, fibroblasts do not express SOD2 and are, therefore, used as negative, internal biological control of the reaction.

The analysis of SOD2 expression was performed in images taken at higher magnification (400x) as described in the section methods.

“There does not appear to be any cross staining to identify the SCC which makes the images in Figure 1 difficult to interpret”

Hematoxylin and eosin-stained sections were used to determine the presence of tumor tissue in our samples before immunohistochemistry assays. Immunohistochemistry was used to assess SOD2 expression and the staining was analyzed in tumor tissue. Tumor tissue (SCC) and stroma (S) are marked in figure 1.

“the manuscript as a whole is way to verbose and includes much too much basic information, e.g. the entire paragraph at the top of page 6 of 12 is a very very basic overview of SOD2 and ROS and it is not necessary.”

The paragraph was removed.

Reviewer 2 Report

The manuscript antioxidants-1179781, “High Expression of SOD2 Protein Is a Strong Prognostic Factor for Stage IIIB Squamous Cell Cervical Carcinoma”, is a fairly well written manuscripts that describes the expression level of superoxide dismutase 2 (SOD2) in cervical cancer. The authors postulate that SOD2 may be used as a biomarker to predict/measure/detect cervical cancer at various stages.  Based on their findings, they conclude “High SOD2 expression is a strong prognostic factor for stage IIIB squamous cell carcinoma of the cervix. SOD2 expression could be an aspirant for biological marker in clinical approaches of women with cervical carcinoma, also a therapeutic target for malignant neoplasms.“  Although the conclusion is intriguing, I don’t feel that there is a clear path to that conclusion based on the data for the following reasons.

The authors admit to the small sample size.  I feel that the size was decent compared to other reports, it would be too small to detect any changes across a time continuum.  Increasing the sample size would be important.

What are other reasons that SOD2 may be increased?  Would these changes be unique to cervical cancer?  There are many other disorders that result in an increased oxidative stress, and an increase in SOD2 expression.  There would need to be something a bit more unique about SOD2 for it to be a potential biomarker (unless it was being biopsied in cervical tissue – then it would be specific to the cervix).

There should be more discussion on SOD2 in general and certainly more specifically as it relates to cancers.  This will increase the readership of the article and give it a broad appeal.  Currently, it is related to stage IIIB – very specific and focused. 

More discussion on what is known of SOD2 activity would be good.  We can have elevated protein expression, but we don’t know if it is a functional protein, or if the machinery is present and functioning to process SOD2 properly to its functional form.

The data analysis, presentation and the imagining is good and appropriate for the design of the study.  There are no changes there.

Below are some minor changes/suggestions for consideration, with the line # before the statement.

36 – insert ‘a’ before biological

46 – delete ‘thus’

47 – change ‘Currently, HPV testing, cytology, and HPV testing and cotesting using cytology are’ to ‘Currently, HPV testing, cytology, or a combination of HPV testing cytology are’

54-56 – for the sentence “For prediction, it could be used to estimate a patient's risk of developing the disease, to screen cancers in routine exams, and to determine a tumor's malignancy.” The word ‘to’ is used repetitively.  Could be:

“For prediction, SOD2 expression could be used to estimate a patient's risk of developing the disease, screen cancers in routine exams, and determine a tumor's malignancy.”

64 – change ‘which’ to ‘that’

75 – hyphenate ‘early-stage’

93 – delete ‘a’ before ‘high-dose’

98 – change ‘month’s interval’ to ‘month intervals’

101 – hyphenate ‘paraffin-embedded’

115 – should ‘Mannhein’ be ‘Mannheim’?

118 – delete ‘a’ before cervical

124 – insert ‘the’ before ‘total’

131 – hyphenate ‘open-source’

132 – delete ‘of’ before ‘the total number’

137 – insert ‘a’ before continuous

138 – change “nonstatistically significant difference was observed (table S2)” to ‘no difference was observed (Table S2)’

155 – delete the second ‘of’ in the sentence.

196 – change ‘is able to’ to ‘can’

206 – delete ‘clearly’

218 – insert ‘the’ before primary

224 – delete the 2nd ‘on’ in the sentence

226 – insert ‘the’ before disease – or specify to which disease this refers to.

242 – change ‘In order to prevent’ to ‘To prevent’

246 – delete the 2nd ‘in’ in the sentence

268 – change ‘suggest’ to ‘suggests’

272 – insert ‘a’ before high

281 – insert ‘a’ before biological

295 – insert ‘a’ before predictor (same in line 299)

Author Response

Response to Reviewer’s Comments

Dear Editor and Reviewers,

Coauthors and I very much appreciated these encouraging, critical and constructive comments and suggestions provided by reviewers on this manuscript, and we strongly believe these have contributed to increase the scientific value and to communicate more clearly. We are submitting the corrected manuscript with changes incorporated the manuscript. These changes aim to address comments given by reviewers. Additionally, we have added notes below to each comment provided:

Once again, we thank you for your time spent to review our paper.

We look forward to meet your expectations.

-The authors

Reviewer #2:

The manuscript antioxidants-1179781, “High Expression of SOD2 Protein Is a Strong Prognostic Factor for Stage IIIB Squamous Cell Cervical Carcinoma”, is a fairly well written manuscripts that describes the expression level of superoxide dismutase 2 (SOD2) in cervical cancer. The authors postulate that SOD2 may be used as a biomarker to predict/measure/detect cervical cancer at various stages.  Based on their findings, they conclude “High SOD2 expression is a strong prognostic factor for stage IIIB squamous cell carcinoma of the cervix. SOD2 expression could be an aspirant for biological marker in clinical approaches of women with cervical carcinoma, also a therapeutic target for malignant neoplasms.“  Although the conclusion is intriguing, I don’t feel that there is a clear path to that conclusion based on the data for the following reasons.

We wish to deeply thank the Reviewer #2 for the careful and constructive revision of the manuscript. The issues raised by the Reviewer have certainly contributed to improve our work.  Below there is a point-by-point answer to issues raised by the Reviewer.

We thank the reviewer for raising this interesting point, giving us the possibility to discuss it properly. In order to better contextualize our study and accurately describe our results we have added the following statement (lines 36-38): “High SOD2 expression is a strong prognostic factor for stage IIIB squamous cell carcinoma of the cervix. SOD2 expression could be used as a prognostic marker in women with cervical carcinoma.”  Besides, the following statement (lines 256-257) was added in the Discussion: “Our results raise the possibility of using SOD2 as a therapeutic target for malignant neoplasms.”

“The authors admit to the small sample size.  I feel that the size was decent compared to other reports, it would be too small to detect any changes across a time continuum.  Increasing the sample size would be important.”

We appreciate the comment. Increasing the sample size, including the use of samples from other centers, is part of our plan for future studies. 

What are other reasons that SOD2 may be increased?  Would these changes be unique to cervical cancer? “There are many other disorders that result in an increased oxidative stress, and an increase in SOD2 expression.  There would need to be something a bit more unique about SOD2 for it to be a potential biomarker (unless it was being biopsied in cervical tissue – then it would be specific to the cervix).”

We thank Reviewer #2 for the opportunity of discussing this interesting point. As shown in references 30-43 of our manuscript, altered SOD2 expression is a common feature of different tumor types and, in most cases, is associated with poor prognosis. Our results show that high SOD2 expression is associated with reduced disease-free survival (DFS) and overall survival (OS) in patients with stage IIIB squamous cell carcinoma of the cervix. At present, we do not know the molecular mechanisms underlying the induction of SOD2 expression in our samples. However, considering the results presented in this manuscript, SOD2 levels are associated with cervical disease outcome. This allow us to speculate that it may be considered a biomarker for cervical cancer. Besides, it is possible that, after further confirmations of our results, knowing the levels of SOD2 may help the clinicians to decide for specific therapeutic approaches in the future.

Several factors may affect SOD2 expression in different settings, including inflammation, oxidative stress and altered cell metabolism. One central molecule involved in SOD2 activation is the transcription factor NF-kB. Considering cervical cancer and its association with HPV infection, several reports have shown that HPV-transformed cells or cells expressing HPV oncogenes exhibit altered activation of NF-kB signaling pathway. We have previously reported that NF-kB activation was associated with SOD2 induction in HPV-immortalized cells (Termini et al. 2008). Major alterations in NF-kB signaling pathway have also been reported by other (Cabeça et al. 2019; Nees et al. 2001).   Since alterations in NF-kB activation are not unique to cervical cancer, we can anticipate that altered SOD2 expression in other, non-HPV related, tumors may be associated with dysregulated activation of this factor.

The current available data does not allow us to describe the molecular pathway underlying SOD2 upregulation in a subset of cervical cancers. We are conducting some functional experiments to dissect the molecular events involved in SOD2 regulation in cells expressing HPV oncogenes and to translate these observations to clinical samples. We hope will be able to present our results in a new manuscript in the future.

To address the concern raised by Reviewer #2 and to better contextualize our results we have added the following statement to the Discussion (lines 269-274): “The observations described above indicate that increased SOD2 expression is a common trait of different tumor types. However, the results presented in our study clearly indicate that the levels of expression of this protein are associated with disease-free survival (DFS) and overall survival (OS) in patients with stage IIIB squamous cell carcinoma of the cervix, suggesting the SOD2 expression may be considered a biomarker for cervical disease outcome.

Termini L, Boccardo E, Esteves GH, Hirata R Jr, Martins WK, Colo AE, Neves EJ, Villa LL, Reis LF. Characterization of global transcription profile of normal and HPV-immortalized keratinocytes and their response to TNF treatment. BMC Med Genomics. 2008 Jun 27;1:29. doi: 10.1186/1755-8794-1-29. PMID: 18588690; PMCID: PMC2459201.

Cabeça TK, de Mello Abreu A, Andrette R, de Souza Lino V, Morale MG, Aguayo F, Termini L, Villa LL, Lepique AP, Boccardo E. HPV-Mediated Resistance to TNF and TRAIL Is Characterized by Global Alterations in Apoptosis Regulatory Factors, Dysregulation of Death Receptors, and Induction of ROS/RNS. Int J Mol Sci. 2019 Jan 8;20(1):198. doi: 10.3390/ijms20010198. PMID: 30625987; PMCID: PMC6337392.

Nees M, Geoghegan JM, Hyman T, Frank S, Miller L, Woodworth CD. Papillomavirus type 16 oncogenes downregulate expression of interferon-responsive genes and upregulate proliferation-associated and NF-kappaB-responsive genes in cervical keratinocytes. J Virol. 2001 May;75(9):4283-96. doi: 10.1128/JVI.75.9.4283-4296.2001. PMID: 11287578; PMCID: PMC114174.

“There should be more discussion on SOD2 in general and certainly more specifically as it relates to cancers.  This will increase the readership of the article and give it a broad appeal.  Currently, it is related to stage IIIB – very specific and focused. 

More discussion on what is known of SOD2 activity would be good.  We can have elevated protein expression, but we don’t know if it is a functional protein, or if the machinery is present and functioning to process SOD2 properly to its functional form.”

This is a very interesting suggestion. Although the goal of our research was not to give an extensive description of SOD2 regulation in cancer we agree with Reviewer #2 that some additional information would improve the Discussion. Therefore, we added the following statements (lines 221-223): “Studies conducted during the last decades indicate that SOD2 plays a dual role in cancer. This protein may exhibit both anti-tumoral as well as pro-tumoral roles depending on tumor type, tumor stage and cellular context.” and (245-252): “The regulation of SOD2 expression and activity occurs both at the transcriptional and posttranslational levels. The activity of SOD2 protein can be regulated by protein localization, interaction with other cellular factors, transition metal incorporation, oxidation, nitration, S-Glutathionylation, phosphorylation, ubiquitination, and acetylation. An extensive in-depth discussion of the complex regulation of SOD2 expression and activity in cancer is beyond the scope of our study. For a detailed description of the many variables involved in SOD2 regulation in cancer the reader is referred to several recent excellent reviews (21,25,27).

“The data analysis, presentation and the imagining is good and appropriate for the design of the study.  There are no changes there.”

We thank Reviewer #2.

Below are some minor changes/suggestions for consideration, with the line # before the statement.

Again, we thank Reviewer #2 for the careful reading of the manuscript. The text has been revised thoroughly to correct errors and all the suggestions made by Reviewer #2 were incorporated to the revised version of manuscript.

Round 2

Reviewer 1 Report

The authors have done an excellent job in addressing the comments, including highlighting why they disagree in some cases. The only remaining comment is that it would be beneficial if the authors were to add reference to the manuscripts by Vijayashree et al, Da Costa et al, and Barreta et al to the manuscript for ease of reference to readers interested in the methodology.

Author Response

“The authors have done an excellent job in addressing the comments, including highlighting why they disagree in some cases. The only remaining comment is that it would be beneficial if the authors were to add reference to the manuscripts by Vijayashree et al, Da Costa et al, and Barreta et al to the manuscript for ease of reference to readers interested in the methodology.”

We thank Reviewer #1 for this encouraging commentary. Following Reviewer #1 advice, the suggested references were added in the “Material and Methods” section, (lines 130 and 143).

Finally, the text has been revised thoroughly to correct errors.